# Alternative Invasion Mechanisms and Host Immune Response to *Plasmodium vivax* Malaria: Trends and Future Directions

**DOI:** 10.3390/microorganisms9010015

**Published:** 2020-12-23

**Authors:** Daniel Kepple, Kareen Pestana, Junya Tomida, Abnet Abebe, Lemu Golassa, Eugenia Lo

**Affiliations:** 1Biological Sciences, University of North Carolina, Charlotte, NC 28223, USA; Kpestana@uncc.edu (K.P.); Junya.tomida@uncc.edu (J.T.); 2Ethiopian Public Health Institute, Addis Ababa 1000, Ethiopia; Abnetabas@gmail.com; 3Aklilu Lemma Institute of Pathobiology, Addis Ababa University, Addis Ababa 1000, Ethiopia; Lgolassa@gmail.com

**Keywords:** *Plasmodium vivax*, erythrocyte invasion mechanisms, Duffy negative, Africa, immunology, epidemiology

## Abstract

*Plasmodium vivax* malaria is a neglected tropical disease, despite being more geographically widespread than any other form of malaria. The documentation of *P. vivax* infections in different parts of Africa where Duffy-negative individuals are predominant suggested that there are alternative pathways for *P. vivax* to invade human erythrocytes. Duffy-negative individuals may be just as fit as Duffy-positive individuals and are no longer resistant to *P.*
*vivax* malaria. In this review, we describe the complexity of *P. vivax* malaria, characterize pathogenesis and candidate invasion genes of *P. vivax*, and host immune responses to *P. vivax* infections. We provide a comprehensive review on parasite ligands in several *Plasmodium* species that further justify candidate genes in *P. vivax*. We also summarize previous genomic and transcriptomic studies related to the identification of ligand and receptor proteins in *P. vivax* erythrocyte invasion. Finally, we identify topics that remain unclear and propose future studies that will greatly contribute to our knowledge of *P. vivax*.

## 1. Introduction

*Plasmodium vivax* malaria is a neglected tropical disease, despite being more geographically widespread than any other form of malaria [1], and causes 132–391 million clinical infections each year [2]. Compared to *P. falciparum*, *P. vivax* has a broader temperature tolerance and an earlier onset of gametocyte development, enabling the parasites to spread through diverse climates [3] and making them more difficult to control and eliminate [4]. Currently, there is no vaccine available for *P. vivax*, though several preventative medications have been shown to be effective [5,6]. The epidemiology of *P. vivax* malaria is further complicated by the pathogen’s unique ability to form dormant-stage hypnozoites in the host liver cells, giving rise to recurrent relapse infections from weeks/months to years later [7,8]. Relapse infections have substantially impacted progress in malaria control, especially in countries that are approaching elimination [9,10,11].

*Plasmodium vivax* was previously thought to be rare or absent in Africa because people of African descent often lack the expression of a Duffy blood group antigen, known as the Duffy antigen–chemokine receptor (DARC). It is believed that the fixation of the Duffy negativity trait, and the rarity *of P. vivax* infection in Africa supports that Duffy-negative individuals are refractory to *P. vivax.* Unlike *P. falciparum*, which utilizes multiple erythrocyte receptors for invasion and has merozoite proteins with overlapping and redundant receptor-binding functions, invasion of erythrocyte by *P. vivax* merozoites exclusively relies on the interaction between PvDBP and DARC expressed on the surface of erythrocytes and reticulocytes. DARC is a glycoprotein on the surface of red blood cells (RBCs) that allows *P. vivax* to bind and invade human erythrocytes at the cysteine-rich region II of Duffy Binding Protein 1 (DBP1) [12,13,14]. However, recent studies have reported several cases of *P. vivax* in Duffy-negative people in different parts of Africa where Duffy-negative populations are predominant [15,16,17]. It is apparent that Duffy-negative individuals are no longer resistant to *P. vivax* malaria [16,17]. This phenomenon raises important questions of how *P. vivax* invades erythrocytes of Duffy-negative individuals. To date, only a single *P. vivax* ligand protein PvDBP1 has been studied in great detail [17]. It has been hypothesized that either mutations in *PvDBP*1 provided a new pathway of entry, or a low expression of DARC in Duffy-negative individuals binds readily with parasites that contain high copies of *PvDBP*1 [18,19]. Recent studies have shown that, despite several mutational differences observed in *PvDBP*1 between Duffy-positive and Duffy-negative infections, none of them bind to Duffy-negative erythrocytes [17], implying that an alternative parasite ligand is being used.

The investigation of erythrocyte invasion mechanisms in *P. vivax* could be complicated by the genetic characteristics and epidemiology of *P. vivax* in Duffy-negative individuals. *P. vivax* has a significantly higher nucleotide diversity at the genome level, compared to *P. falciparum* [20]. Such a contrast could be attributed to frequent gene flow via human movement, intense transmission, and variation in host susceptibility [21,22,23]. Genes associated with erythrocyte binding, such as Duffy binding protein (*PvDBP*), erythrocyte binding protein (*PvEBP*), reticulocyte binding protein (*PvRBP*), merozoite surface protein (*PvMSP*), apical membrane antigen 1 (*PvAMA*1), and tryptophan-rich antigen genes (*PvTRAg*) families, are highly diverse in *P. vivax* from Africa and Southeast Asia [24,25,26,27,28]. These genes have been shown to play a role in reticulocyte invasion [24,28] and patient antigenicity [29,30] and provide explanations to high levels of selection detected at the genome levels in *P. vivax* from South Korea [31], Kyrgyz Republic [32], New Guinea [33], and Thailand [34]. Proteins such as RBP, TRAg, anchored micronemal antigen (GAMA), and Rhoptry neck protein (RON) have been suggested to play a role in red cell invasion, especially in low-density infections [35,36,37,38,39]. Unfortunately, studies that investigated erythrocyte invasion pathways are scattered with no definitive evidence and systematic approaches to clarify the exact role of these target genes. Due to a lack of reliable and logistical long-term in vitro methods [40], *P. vivax* remains a parasite for which it is difficult to effectively study the molecular mechanisms and biology in detail, beyond genetic characterizations.

In this paper, we present a systematic review of the literature, to highlight the complexity of *P. vivax* malaria and characterize *P. vivax* candidate invasion genes, pathogenesis, and host immune responses. We provide a comprehensive review on parasite ligands in several *Plasmodium* species that further justify candidate genes in *P. vivax*. We also summarize previous genomic and transcriptomic studies related to the identification of ligand and receptor proteins in *P. vivax* invasion. Finally, we identify topics that remain unclear and propose future studies that will greatly contribute to our knowledge of *P. vivax*.

## 2. Pathogenesis of *P. vivax*


Recent findings of *P. vivax* cases in Duffy-negative individuals suggest that some lineages may have evolved to use ligands other than Duffy for erythrocyte invasion [17]. This significantly increases the risk of *P. vivax* infection in the African populations and may eventually become a new cause of epidemics and severe disease across Africa. To establish how the phenomenon of *P. vivax* infection of Duffy-negative individuals has evolved and identify potential vaccine candidates to target it, it is important to understand how this parasite invades Duffy-negative erythrocytes and, hence, causes malaria. The investigations of *P. vivax* at the cellular and molecular levels have been restricted by the lack of a continuous in vitro culturing of live parasites. With the advancement in *P. vivax* genome sequencing technology, coupled with the ability to mature ex vivo isolates, it is now possible to obtain high-quality transcriptomes of the blood stages. However, there is still a lack of viable methods to indefinitely culture *P. vivax*, due to the need for young reticulocytes to sustain long-term culture. Strategies to overcoming this problem have been proposed but remain impractical due to a large initial and continuous investment of labor and infrastructure [41]. The successes of short-term culture utilizing young reticulocytes from placental blood [40,42] and indefinite culture in *Saimiri boliviensis* and *Aotus nancymae* monkeys [43,44] shed light on pathogenesis in humans and potential ligands for invasion [39,44]; however, several unanswered questions remain.

While mature asexual *P. vivax* and its transmissive gametocytes occur in peripheral blood, histological analyses of *P. vivax* in *Aotus* and *Saimiri* monkeys have shown immature gametocytes and few asexual schizonts present in the parenchyma of bone marrow [45]. Asexual schizonts appear to be more concentrated in the sinusoids of the liver [45], suggesting that bone marrow could be a critical reservoir for *P. vivax* gametocyte development and proliferation. Indeed, the bone marrow reservoirs may suggest that microscopic detection is not ideal for active case detection and treatment of *P. vivax* until bone marrow samples are accessible. As *P. vivax* requires reticulocytes for growth [46,47,48,49], the general low proportion of reticulocytes (that make up only 1% of the total number of host erythrocytes) may explain low parasite loads in symptomatic patients [50,51,52] and a lack of observable schizonts in blood circulation [52,53]. Additionally, pathological analyses of *S. boliviensis* tissues showed that *P. vivax* infections also affect the lungs and kidneys, both of which had mononuclear infiltrates, higher macrophage levels, alveolar wall thickening, collagen deposition, and type II pneumocyte hyperplasia [44]. The level of tissue damage is parasite-load dependent and determined by the amount of by-product, namely hemozoin, being produced [44]. These findings may imply a large number of asymptomatic *P. vivax* carriers in the general populations. It is well-known that *P. vivax* has the ability to relapse from dormant liver-stage hypnozoites, from weeks to years after the clearance of the primary blood-stage infection, and this is a major obstacle to its control and elimination [20,54]. The liver and bone marrow have been shown to be major parasite reservoirs for *P. vivax* hypnozoites in *Saimiri* monkey models [45,55], but mechanisms of hypnozoite development remain largely unknown and are difficult to study due to a lack of long-term in vitro culture. Moreover, relapse varies systemically by geographic region and/or seasonal changes in the environment [54]. In regions where *P. vivax* transmission is intense and stable, relapse is common and enhances local transmission [20,54], whereas, in Africa, *P. vivax* transmission is relatively low and usually seasonal and unstable [56,57,58]. The rate of relapse is largely unknown. There is, as of yet, no information on the frequency and clinical impacts of relapse in Duffy-negative *P. vivax* infections, nor reliable biomarkers for relapse detection, due to limited technologies and substantial knowledge gaps in the biology of *P. vivax* hypnozoites and relapse. Future investigations employing a longitudinal study design that monitors the dynamics and consequences of relapse infections in both Duffy-positive and Duffy-negative individuals will offer deep insights into the epidemiology and biology of *P. vivax* infections.

## 3. Erythrocyte Invasion Mechanisms in Non-*Plasmodium vivax*

### 3.1. Plasmodium falciparum

Our current knowledge of the molecular mechanisms of erythrocyte invasion in several *Plasmodium* species offers a reference model on candidate invasion ligands in *P. vivax*. *Plasmodium falciparum* invades a wide range of red blood cells, from young reticulocytes to mature normocytes. One of the main binding protein ligands is the erythrocyte binding ligand (EBL) family, which includes multiple members, such as EBA-175, EBA-140, EBL-1, and EBA-181. EBA-175 binds to the sialic acid–containing structure on human erythrocyte receptor glycophorin A (GpA) during invasion [59]. The role of the EBA-175 protein has been shown to be critical for erythrocyte invasion, as antibodies raised against EBA-175 prevent binding to GpA in vitro [60,61]. EBA-175 triggers changes in the erythrocyte membrane [62,63], and the shedding of EBA-175 causes uninfected red blood cells to cluster or form rosette, which allows for immune evasion [64]. The host immune responses may explain the polymorphisms and diversifying selection observed in EBA-175 [65]. Other ligands, such as EBA-140 and EBL-1, are known to bind to glycophorin C (GpC) [66] and glycophorin B (GpB), respectively, on the erythrocytes. Unlike GpA and GpC, the GpB exhibits high levels of polymorphisms, particularly in people of African ancestry, suggesting that a strong selective pressure may have provided an evolutionary advantage to parasite invasion [67]. For example, the S-s-U- and Dantu GpB phenotypes both showed moderate protection against invasion; however, this does not hold true for all GpB phenotypes [67,68,69,70]. To the best of our knowledge, the specific receptor for EBA-181 is chymotrypsin-sensitive, trypsin-resistant, and neuraminidase-sensitive to erythrocytic treatment [71,72], although it remains to be identified. 

Another important binding protein family of *P. falciparum* is the reticulocyte-binding homologue (PfRh) that includes PfRh1, PfRh2a, PfRh2b, PfRh4, and PfRh5. PfRh1 binds to an unidentified receptor “Y”, which has been characterized to be trypsin- and chymotrypsin-resistant and neuraminidase-sensitive [73,74]. PfRh1 is necessary for sialic acid–dependent invasion of human red blood cells [74]. Antibodies raised against PfRh1 have been shown to block invasion by inhibiting calcium signaling in the merozoite [75]. PfRh2a and PfRh2b are identical for much of the N-terminus region, but each has a unique 500 C-terminus region [76] and differential expressions in various *P. falciparum* lines, including deletions, such as a deletion of PfRh2b in *P. falciparum* D10 [76,77]. The loss of PfRh2b does not appear to impact invasion or growth of the parasites and suggests compensatory mechanisms for the loss of PfRh2b [78]. PfRh2a binds to more than one receptor on erythrocytes, but these receptors have yet to be identified [79,80]. PfRh2b has been shown to be involved in merozoite calcium signaling [80]. It binds to an unknown receptor “Z” on erythrocytes, which is neuraminidase- and trypsin-resistant and chymotrypsin-sensitive [81]. PfRh4 has been shown to have sialic acid–independent binding activity with the complement receptor type I (CR1) on erythrocytes [82,83]. The PfRh5 complex is composed of PfRh5, Ripr, CyRPA, and Pf113, which collectively promote successful merozoite invasion of erythrocytes by binding to basigin (BSG, CD147) [84,85]. A BSG variant on erythrocytes, known as Ok^a-^, has been shown to reduce merozoite binding affinities and invasion efficiencies [86]. This variant was reported so far only from people of Japanese ancestry [87]. Previous knockout or double-knockout experiments have indicated that the *EBL* and *PfRh* gene families work cooperatively or can functionally compensate for the loss of each other [88,89]. For example, a loss of *EBA*-175 can activate *PfRh*4 [88,89]. When *EBA*-181 expression was disrupted, *PfRH*2b was no longer functional [89]. When *EBA*-181 and *EBA*-140 genes were disrupted, the parasite deleted the *PfRh2b* gene [89]. Further study is needed to gain a deeper understanding of how they may work synergistically to promote invasion and immune evasion.

### 3.2. Plasmodium knowlesi

Until recently, *P. knowlesi* was considered primarily a simian malaria that infects *Macaca fascicularis*, *Macaca nemestrina,* and *Presbystis melalophos* [90]. *P. knowlesi* is now confirmed to cause malarial infections in humans [91]. *P. knowlesi* has been shown to use different ligands to invade macaques and human erythrocytes [90]. Two gene families, *DBL* and *RBP*, are responsible for erythrocyte binding. The *DBL* gene family comprises *PkDBP*-α, *PkDBP*-β, and *PkDBP*-γ. In humans, the parasite ligand responsible for erythrocyte invasion is PkDBP-α, which binds to the DARC receptor. The other two Duffy-binding proteins, PkDBP-β and PkDBP-γ, bind only to macaque but not human erythrocytes [14]. The normocyte-binding protein Xa (NBPXa) is required for binding in human erythrocytes, but it is not necessary for invasion of *Macaca mulatta* erythrocytes [92]. Variation in PkNBPXa has been shown to be linked with parasite virulence and severity of disease [93]. The receptors for NBPXa and NBPXb necessary for invasion for either human or *M. mulatta* erythrocytes have yet to be identified [90]. Unlike *P. vivax*, both *P. falciparum* and *P. knowlesi* can be maintained in long-term culture, making them ideal systems for studying invasion mechanisms [94,95].

### 3.3. Plasmodium cynomolgi

*P. cynomolgi* is a vivax-like simian malaria that shares many genomic and phenotypic characteristics with *P. vivax* and has been often used as a reference model of *P. vivax* [96]. Two gene families, erythrocyte binding-like (*EBL*) and reticulocyte binding-like (*RBL*), are responsible for erythrocyte binding and invasion in *P. cynomolgi* [97,98,99]. The *EBL* gene family encodes *PcyDBP*-1 and *PcyDBP*-2, similar to *PkDBP,* which binds to the complementary DARC receptor on Duffy-positive erythrocytes. *PcyDBP-1* is an ortholog for *PkDBP*-α, while *PcyDBP*-2 has no known orthologs with other *Plasmodium DBP*s [100]. Previous studies have shown no variation in gene copy number of either *PcyDBP-1* or *PcyDBP-2* among *P. cynomolgi* laboratory strains [101]. Studies of field isolates of both *P. cynomolgi* and *P. knowlesi* have shown that *PcyDBP-1* exhibits high levels of nucleotide diversity, as compared to *PcyDBP2* or *PkDBP*s [102]. The *RBL* gene family is composed of *PcyRBP*1, *PcyRBP*1a, *PcyRBP*1b, *PcyRBP*2a, *PcyRBP*2b, *PcyRBP*2c, *PcyRBP*2d, *PcyRBP*2e, *PcyRBP*2f, and *PcyRBP*3, most of which are responsible for mediating parasite invasion into reticulocytes [101]. Functional studies of *PcyRBP*s are further complicated, as different strains of *P. cynomolgi* have a different set of *RBL* genes. For example, *PcyRBP2a* is present in the *P. cynomolgi* B and *P. cynomolgi* Cambodian strains but absent in the *P. cynomolgi* Berok strain. Similarly, *PcyRBP*1b is present in the *P. cynomolgi* Berok and Gombak strain but absent in the *P. cynomolgi* B, Cambodian and Rossan strain [96]. While it is possible that the loss of *PcyRBP1b* can be compensated by the presence of *PcyRBP2a* [101], the relative role of *PcyRBP1b* and *PcyRBP2a* in RBC invasion requires further investigations [103].

## 4. *Plasmodium vivax* Ligand Proteins and Host Receptors for Erythrocyte Invasion

The Duffy-binding protein of *P. vivax* (PvDBP) and *P. knowlesi* (PkDBP-α) interact with DARC on erythrocytes [104]. A 140 kD region II of PvDBP (amino acids at site 198–522; Table 1) was identified as the key binding site to human erythrocytes [14]. In humans, there are two major codominant alleles for DARC, Fy^a^ and Fy^b^, which differ by a single nucleotide substitution at amino acid position 42 with glycine and aspartic acid, respectively. Individuals who are Fy^a+b-^ have been shown to be at reduced risk for clinical *P. vivax* in comparison to Fy^a-b+^ individuals [105,106]. Individuals with a single point mutation c.1-67T>C (rs2814778) in the GATA-1 box of the *DARC* gene are considered Duffy-negative (Fy^-^), as erythrocytic expression of DARC is abolished. Duffy-negative (Fy^-^Fy^-^) individuals were previously thought to be resistant to infection by *P. vivax* and *P. knowlesi* due to the parasites’ inability to infect erythrocytes [107], but several recent studies have shown that *P. vivax* can infect Duffy-negative individuals, potentially utilizing another invasion ligand(s) [108]. The *P. vivax* erythrocyte binding protein (PvEBP), which is similar to PcyM DBP2 sequences in *P. cynomolgi* and contains a Duffy-binding like domain, was discovered in 2013 by de novo genome assembly of field isolates from Cambodia [109]. Binding assay of PvEBP region II (amino acids at site 171-484; Table 1) showed that, unlike PvDBP region II, PvEBP is able to moderately bind to Duffy-negative erythrocytes [17], lending support to the hypothesis of an alternative invasion pathway in *P. vivax*. 

The comparisons of genomic sequences between *P. vivax* and other *Plasmodium* species have identified several members of the *P. vivax* reticulocyte binding protein (RBP) gene family [47,110,111]. The *P. vivax* RBP gene family comprises several full-length genes, including *PvRBP1a*, *PvRBP1b*, *PvRBP2a*, *PvRBP2b*, and *PvRBP2c*; partial genes, including *PvRBP1p1*, *PvRBP2p1*, and *PvRBP2p2*; and pseudogenes, including *PvRBP2d*, *PvRBP2e*, and *PvRBP3*. *PvRBP1* comprises *PvRBP1a* and *PvRBP1b* and shares homologous regions with *P. falciparum PfR*h4 [112]. The binding regions of *PvRBP1a* and *PvRBP1b* are homologous to that of *PfRh*4, and the amino acids at site ~339–599 were confirmed to interact with human reticulocytes [83,113] (Table 1). *PvRBP1a* is orthologous to the pseudogene of *PkNBP*1, but no orthologous region of *PvRBP1b* was detected in *P. knowlesi* [96]. While the host receptors of both PvRBP1a and PvRBP1b proteins are unclear, it has been shown that the receptors are neuraminidase resistant [83]. PvRBP1p1 contains a fragment which has 95% similarity to a C-terminal sequence in PvRBP1b [114]. Several members of *PvRBP*2 (*PvRBP*2a, *PvRBP*2b, *PvRBP*2c, *PvRBP*2d, *PvRBP*2e, *PvRBP*2p1, and *PvRBP2*p2) are orthologous to *PfRH*2a and *PfRH*2b [83,110], as well as *PcyRBP*2 [96]. Some of them, such as PvRBP2a and PfRh5, also share high structural similarity [115]. *PvRBP*2b and *PvRBP*2c are orthologous to *PcyRBP*2b and *PcyRBP*2c, respectively [96]. Recently, the receptor for PvRBP2b has been identified as transferrin receptor 1 (TfR1) and the PvRBP2b–TfR1 interaction plays a critical role in reticulocyte invasion in Duffy-positive infections [116]. While PvRBP*2p*1 has been identified in all human *P. vivax* infections, *PvRBP2p*2 was only present in certain lineages of *P. vivax* [117]. *PvRBP*2d, *PvRBP*2e, and *PvRBP*3 are pseudogenes that share homology with other *PvRBP*s but encode for nonfunctional proteins [117]. In addition, *PvRBP2e* is present in the Cambodian field isolates but not in the *P. vivax* Salvador I [109]. The role of these *PvRBP* genes in erythrocyte invasion remains unclear.

Tryptophan-rich antigens (TRAgs) are a family of antigens found on human and rodent malaria parasites. In *P. vivax*, the number of encoded tryptophan-rich proteins is much higher than that of *P. falciparum*. In the latter case, some of these proteins have been shown to play important role in red-cell invasion and thus are proposed as potential vaccine candidates against *P. falciparum* malaria [18,19]. Although the reason as to why such a large number of tryptophan-rich proteins are being expressed by *P. vivax* is unknown, the stage-specific expression of these genes is indicative of their different roles in the parasite’s life cycle [20,118]. The PvTRAg family contains 36 members, each with a positionally conserved tryptophan-rich domain in the C-terminus region [119,120]. Of the 36 PvTRAgs, 33 transcribe differently during the ring, trophozoite, and schizont stages of *P. vivax*, indicating their involvement in blood-stage development [35]. A majority of transcription take place during the schizont-ring transition [121], suggesting their role in erythrocyte invasion. The proportion of non-synonymous SNPs in *PvTRAg* genes was shown to be significantly high, suggesting the effect of diversifying selection related to antigenic function [122]. Nine PvTRAgs have been shown to bind to human erythrocytes, including PvTRAg33.5, PvTRAg35.2, PvTRAg69.4, PvTRAg34, PvTRAg38, PvTRAg36, PvTRAg74, PvTRAg26.3, and PvTRAg36.6. These proteins possess erythrocyte-binding activity with predicted protein localization to be during the schizont stage [123,124], and each protein recognizes multiple erythrocyte receptors [124]. A comparison of *P. vivax* transcriptomes between *Aotus* and *Saimiri* monkeys indicated the expression of six *PvTRAg* genes in *Saimiri P. vivax* was 37-fold higher than in the *Aotus* monkey strains [39]. Five of these highly expressed *PvTRAg* genes were previously shown to bind to human erythrocytes [38,123]. Although most TRAg receptors remain poorly characterized and unnamed, the receptor of PvTRAg38 has been identified as Band 3 that binds to amino acid positions 197–214 [125] (Table 1). The 10 PvTRAg ligands cross-compete with one other and each receptor is shared by more than one TRAg ligand, e.g., PvTRAg38 and PvTRAg74 share a common chymotrypsin-sensitive erythrocyte receptor [123,124]. In addition, the expression of the 10 PvTRAgs varies and is stage-specific, suggesting their differential roles in parasite growth and development. For example, PvTRAg, PvTRAg26.3, PvTRAg36.6, and PvTRAg69.4 are all expressed at the ring and early trophozoite stages of the parasites, indicative of an important role in rosetting; PvTRAg35.2, PvTRAg38, PvTRAg36, and PvTRAg34 are expressed during the schizont and merozoite stages, indicative of invasion properties [124]. Further, PvTRAg36.6 interacts with early transcribed membrane protein (PvETRAMP) to form a protein complex that is apically localized in merozoites, suggesting that this protein is critical for development or maintenance of the parasitophorous vacuole membrane [38]. Recent studies have shown that the PvTRAg35.2 gene sequences were highly conserved in the parasites, and amino acid residues 155–190 and 263–283 were involved in erythrocyte binding [126] (Table 1). PvTRAg35.2 competes with PvTRAg 33.5 and PvTRAg28 and may play a redundant role in erythrocyte invasion [126]. Orthologs of *Pv-fam-a* are present in *P. knowlesi, P. falciparum,* and *P. yoelii,* suggesting that these genes may play a critical role for invasion throughout *Plasmodium* evolution [111]. Other PvTRAgs, such as PvTRAg56.6 and PvTRAg56.2, interact with PvMSP1 and PvMSP7, to stabilize surface protein complexes on merozoites, and are likely not involved in erythrocyte binding [38]. Additionally, PvTRAgs elicit a strong IgG antibody immune response in *P. vivax*–infected individuals, with memory lasting up to 5–12 years after being infected [35]. Nine TRAgs (PvTRAg3, PvTRAg7, PvTRAg13, PvTRAg14, PvTRAg15, PvTRAg18, PvTRAg20, PvTRAg26, and PvTRAg35) showed high IgG positivity and conserved IgG reactivity in three Asian countries with low malaria endemicity [35], demonstrating the universal antigenicity of these TRAg proteins.

Merozoite Surface Proteins (MSPs) are a large family of genes found on the surface of merozoites, and a few members are suggested to be involved in non-DBP1 erythrocyte invasion pathways [59]. MSP1 is a 200 kDa highly conserved antigen that undergoes several cleavage events as invasion occurs [78,87]. MSP1 shows a strong binding affinity between 20 and 150 nM at the 42 and 19 kDa fragment cleavage sites, with high-activity binding peptides (HABPs) clustered close to these two fragments at positions 280–719 and 1060–1599, respectively [59], suggesting its critical role in erythrocyte invasion. MSP1 has the potential to be a vaccine target due to its strong immunogenicity [127], but further research is needed [128,129]. MSP3 transcribes during the trophozoite and schizont stages of *P. vivax* [130] and is highly expressed in *Saimiri* infections [39]. Although it is yet unclear whether MSP3 binds to human erythrocytes, MSP3.3 and MSP3.5 were expressed on the surface of mature schizonts and interact with MSP1, MSP7, and MSP9 [39]. The MSP3 gene family contained RNA expression of 11 members during the trophozoite and schizont stages, hinting at the MSP3 family playing an important role in both maturation and binding [130]. Interestingly, MSP3.7 was detected at the apical end of merozoites, which differentiates probable roles from other MSP3 family members [130]. Additionally, MSP3.11 transcripts were present, but with no corresponding protein being detected, questioning the exact role of this protein [130]. Although the *MSP*7 gene family has not been shown to bind to erythrocytes, it forms a complex with PvTRAg36.6 and PvTRAg56.2 and localizes on the surface, likely assisting in stabilization of the protein complex at the merozoite surface [38]. Several *MSP*7 genes, including *MSP*7C, *MSP*7H, and *MSP*7I, are strong antibody targets and contain high genetic diversity due to frequent positive selection [131]. The MSP9 family also undergoes frequent selection and recombination and forms a co-ligand complex with the 19 kDa fragment of MSP1, but no erythrocytic binding activity was observed [132]. It is possible that MSP9 assists MSP1 in binding to erythrocytes. MSP9 is highly immunogenic at conserved regions 795–808, making it a good vaccine candidate [133,134].

## 5. Humoral Immune Response against *P. vivax* and Vaccine Targets

The severity of malaria infection during the erythrocyte stage of *Plasmodium* depends on various factors, including the following: the location of parasitized red blood cells in the target organs; the local and systemic action of the parasite’s bioactive products; pro-inflammatory cytokine production, as well as innate and adaptive immune system at the cellular levels that involves cytokine and chemokine regulators; and the activation, recruiting, and infiltration of inflammatory cells [136]. During *P. vivax* infection, some individuals can acquire immunity naturally. Such immunity consists of humoral IgG antibodies, cellular cytokines, and proteolytic enzymes production as part of the host response to the pathogen [137,138]. Apart from the invasion capability of *P. vivax*, host immune response to the pathogen is also a key factor for determining parasitemia and pathology. For example, patients with moderate parasitemia in endemic regions of Colombia were shown with cellular immune responses including high IFN-γ and TNF-α levels and a pro-inflammatory cytokine profile in unstable transmission regions [139]. The balance in interleukin (IL)-10/TNF-α rate could prevent increased parasitemia and host pathology [140]. To date, DBP is the primary target antigen for *P. vivax* vaccine development. Serological studies have shown that region II of PvDBP, which *P. vivax* uses to bind to human erythrocytes, induces antibodies against DBP and is naturally immunogenic in people residing in endemic regions, through repeated exposure to the infection [141,142]. However, the naturally acquired neutralizing antibodies against DBP are short-lived, increasing with acute infection, and are strain specific [143,144]. Antibodies from plasma from naturally exposed people and from animals immunized with recombinant Duffy binding protein (rDBP) have blocked the specific interaction between the PvDBP ligand domain in vitro and its receptor on erythrocyte surface [143,144]. Such inhibitory activity was correlated with antibody titers. The DBP1 binding domain is polymorphic, tending to compromise the efficacy of any vaccine associated with strain-specific immunity [145]. While almost all mutations in polymorphic residues did not alter RBC binding [146], such polymorphism has a synergic effect on the antigenic nature of DBP [147]. However, polymorphic SNP variants in the binding domain of PvDBP1 had no effect on the degree of inhibition by anti-DBP monoclonal antibodies. On the contrary, a higher *PvDBP* gene copy number was shown to reduce susceptibility to anti-PvDBP antibody response [148], but not for better invasion of FyA/FyA and FyA/FyB reticulocytes. 

Apart from DBP, the MSP family is responsible of the interaction between merozoites and reticulocytes during the erythrocyte phase. PvMSP1, PvMSP3, and PvMSP9 are potential vaccine candidates, since they are exposed to the immune system and are recognized by antibodies from naturally infected individuals [149]. Cytoadherence assays demonstrated that MSP1_19_ is an essential adhesion molecule used in *P. vivax* invasion to erythrocytes [150] and is immunogenic in people living in areas of unstable malaria transmission in Southeast Asia, Papua New Guinea, and Brazil [151]. Two recombinant polypeptides, rPvMSP_114_ and rPvMSP_120_, from the MSP1 C-terminal region show high binding activity to reticulocytes, but no antibodies against these peptides were detected in immunized *Aotus* monkeys [87]. MSP3 is an abundant ligand on the merozoite surface that is essential for reticulocyte invasion. PvMSP3α block II and the C-terminal region were shown to be highly immunogenic. Individuals living in an endemic region with a high number of previous episodes of malaria were shown to have increased IgG1 and IgG3 anti-PvMSP3α [149,152,153]. Likewise, the PvMSP9 C-terminal and NT domains have also been shown to induce memory T-cell response (higher IFN-γ and IL-4 cytokine production) in individuals living in *P. vivax*–endemic regions of the Brazilian Amazon and Papua New Guinea [153,154] and are specific targets of *P. vivax* vaccine. It is unclear if Duffy-negative individuals acquire high levels of antibodies against the 19-kDa C-terminal region of the *P. vivax* PvMSP1, PvMSP3, and PvMSP9, resulting in a low susceptibility to *P. vivax* infection [106]. 

PvRBP2b was recently shown to bind transferrin receptor 1 of the reticulocytes, through the apical domain and the protease-like domain of TfR1 and the N-terminal region of Tf [116]. RBP2P1 protein was found to be expressed in schizonts and localized at the apical end of the merozoite, and it preferentially binds reticulocytes over normocytes. Monoclonal antibodies raised against PvRBP2b prevent reticulocyte binding and reduce *P*. *vivax* invasion [116]. *P. vivax* malaria patients had higher IgG levels against rRBP2P1 than did naive individuals. Human antibodies to this protein also exhibit erythrocyte binding inhibition and are associated with lower parasitemia [155]. PvRBP1a and PvRBP1b are highly transcribed during the parasite schizont stage [118]. PvRBP1a-34 and PvRBP1b-32 proteins bounded specifically to reticulocytes and showed significantly higher reticulocyte binding activity than normocyte-binding activity [83]. Clinical assays have indicated that PvRBP1_435–777_ is poorly immunogenic, likely because PvRBP1 had multiple promiscuous T-cell epitopes, which did not induce specific genetic restriction [156]. IgG prevalence against the NT region (including the most polymorphic region) of the PvRBP1a and b was intermediate in a population from Thailand [117], but IgG prevalence against the PvRBP1a-34 and PvRBP1b-32 proteins was significantly higher in *P. vivax* patients than healthy individuals in the Republic of Korea [83]. Moreover, the highly conserved region III (between amino acids 1941–2229), with the greatest amount of high-affinity reticulocyte-binding peptides and high binding affinity, was shown to induce high antibody titers in *Aotus* monkeys and be able to recognize the full PvRBP1 in parasite lysate [157]. The expression of PvRBPs in the African *P. vivax* and the antibody response against PvRBP in Duffy-negatives will provide important implications to the usefulness of a future vaccine in vivax malaria control in Africa.

While the circumsporozoite protein (CSP) is one of the most important proteins described in hepatocyte invasion by *Plasmodium* sporozoites, previous studies involving individuals residing in *P. vivax* malaria–endemic regions in Brazil showed low responses for antibodies directed against the repeat and C-terminal regions [158]. On the other hand, members of the *TRAg* gene family that were highly expressed in non-DARC *Saimiri*-infected *P. vivax* have also shown to induce antibody response in people from vivax-endemic regions. A recent study of 383 children in Papua New Guinea showed that antibodies against PvFAM-D2 were significantly more common in children with active *P. vivax* infections [159]. The coexpression of PvFAM-A2 and PvFAM-D2 proteins in infected reticulocytes is spleen-dependent, based on the *Aotus* monkey model [160]. These proteins were recognized by a high percentage of sera and are highly immunogenic targets of naturally acquired immune responses [160]. These results are in agreement with members of these families being highly expressed in transcriptional analysis of parasite isolates [39,161]. Moreover, PVX_108770 (VIR14) of the multi-gene VIR family largely located at the subtelomeric regions also presented high sero-positivity, despite the role of its conserved globular domains in eliciting cross-reacting antibodies being unclear [162,163]. Other proteins, such as HYP, which is 100% conserved among *P. vivax* isolates from Mauritania, North Korea, India, and Brazil, and had antibodies significantly associated with protection against clinical *P. vivax* episodes in children [159], are a potential target of blood-stage vaccine. Taken together, it is possible that individuals with low-to-no DARC expression have lower susceptibility to infection than individuals having high DARC expression by eliciting high frequency and magnitude of anti-DBP, anti-MSP, anti-RBP, and anti-FAM antibody response against *P. vivax* during the blood stage. This may imply that one of *P. vivax*’s primary mechanisms for evading host immunity works through indirect, negative regulation of DARC, influencing the humoral response against erythrocyte invasion and parasite development.

## 6. Conclusions

The documentation of *P. vivax* infections in different parts of Africa where Duffy-negative individuals are predominant [164,165,166,167,168,169,170,171] suggested that there are alternative pathways for erythrocyte invasion. It is apparent that Duffy-negative individuals are no longer resistant to *P. vivax* malaria. The increased risk of *P. vivax* infection and the growing clinical burden across Africa, as well as in Duffy-negative individuals, certainly highlight the public health concern of *P. vivax* malaria. Future studies should clarify the function and immunogenicity of various candidate parasite ligand proteins and identify their corresponding receptors involved in alternative Duffy-independent erythrocyte invasion, critically examine the host antibody response with respect to *P. vivax* proteins across wide ethic groups, provide a rigid comparison and analysis of asymptomatic reservoirs and transmission mechanisms of *P. vivax* in Duffy-negative populations, and unveil the biological features of relapse infections in Africa. Furthermore, future studies should investigate the biology of relapse infections, including the development and characteristics of hypnozoites in *P. vivax*, which remain largely unclear, despite the overwhelming importance with regard to the impact of relapses on genetic variation of the parasites, transmission, and antimalarial treatment of *P. vivax* cases. Last but not least, technical studies examining feasible methods for indefinite in vitro culture of *P. vivax* are vital for further understanding beyond the “omics” level of the parasites.

## Figures and Tables

**Table 1 microorganisms-09-00015-t001:** Currently known *P. vivax* genes and amino acid regions responsible for binding human erythrocytes.

Gene Name	Amino Acid Binding Region	Target Cell(s)
PvDBP1	198–522 [14]	Duffy-positive erythrocytes
PvEBP/DBP2	171–484 [17]	Erythrocytes
PvRBP1a	352–599 [113]	Reticulocytes
PvRBP1b	339–587 [83]	Reticulocytes
PvRBP2a	160–1000 [115]	Mature RBCs and reticulocytes
PvRBP2b	161–1454 [116]	Reticulocytes
PvRBP2c	Native protein [47]	Reticulocytes
-	168–524 [135]	10% reticulocytes
PvTRAg38 PvTRAg35.2 PvMSP1	464–876 [135]	34% reticulocytes
	197–214 [125]	Erythrocytes
	155–190 [126]	Erythrocytes
	263–283 [126]	Erythrocytes
	280–719 [59]	Erythrocytes
	1060–1599 [59]	Erythrocytes

The empty cells represent the same gene above.

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
