# Peer review of "Alternative Invasion Mechanisms and Host Immune Response to Plasmodium vivax Malaria: Trends and Future Directions"

_microorganisms, 2020, doi:10.3390/microorganisms9010015_

Round 1

Reviewer 1 Report

This is a well written review that discusses several critical aspects of the biology of P. vivax, and how they relate to erythrocyte invasion and host immune response.  The documentation of P. vivax infection in Duffy negative individuals has led many suggestions of alternative pathways of erythrocyte invasion. The authors of this review have put out a lot of very detailed information about P. vivax, while also providing comparative data on erythrocyte invasion in other non-Plasmodium vivax species, which I found to be very useful.

Recognizing the limitation in available data, the one aspect that was not covered in detail are hypnozoites, the liver stage of P. vivax infections, which are particularly important as they are responsible for the multiple relapses post infection.  The authors should have at least articulated what is known and the state of the knowledge. However, this is not a major issue and can be incorporated if the authors want to.

Overall, this is a well written review paper and I highly laud the authors for such a nicely done work.

Reviewer 2 Report

The review article is well written by Keppel et.al, and it arouses valuable questions and great ideas about the mechanisms of malaria parasite invasion. In addition author compared the invasion machinery of both falciparum and vivax parasites, which would be good for the reader.

I have one question:

Author written in the abstract and Introduction part "Finally, we identify topics that remain unclear and propose future studies that will greatly contribute to our knowledge of P. vivax" . Can you list the topic in this article that what you find and its future prospects?
